# Loss of SPRED3 Causes Primary Hypothyroidism and Alters Thyroidal Expression of Autophagy Regulators LC3, p62, and ATG5 in Mice

**DOI:** 10.3390/ijms26157660

**Published:** 2025-08-07

**Authors:** Celine Dogan, Luisa Haas, Rebecca Holzapfel, Franziska Schmitt, Denis Hepbasli, Melanie Ullrich, Michael R. Bösl, Marco Abeßer, Kai Schuh, Sina Gredy

**Affiliations:** 1Institute of Physiology I, University Wuerzburg, Roentgenring 9, 97070 Wuerzburg, Germany; celine.dogan@uni-wuerzburg.de (C.D.); luisa.haas1@stud-mail.uni-wuerzburg.de (L.H.); f.schmitt@daad.de (F.S.); denishepbasli@gmail.com (D.H.); marco.abesser@uni-wuerzburg.de (M.A.); 2Institute of Experimental Biomedicine I, University Hospital Wuerzburg, Rudolf Virchow Center, University of Wuerzburg, 97080 Wuerzburg, Germany; holzapfel_r@ukw.de (R.H.); boesl_m@ukw.de (M.R.B.); 3Center for Rare Diseases, University Clinic Wuerzburg, Josef-Schneider-Strasse 2, 97080 Wuerzburg, Germany; ullrich_m@ukw.de; 4Center for Medical Informatics, University Clinic Wuerzburg, Schweinfurter Strasse 4, 97080 Wuerzburg, Germany

**Keywords:** SPRED, SPRED3, KO mice, hypothyroidism, autophagy

## Abstract

Sprouty-related proteins with enabled/vasodilator-stimulated phosphoprotein homology 1 (EVH1) domain (SPREDs) are negative regulators of the Ras/MAPK signaling pathway and are known to modulate developmental and endocrine processes. While the roles of SPRED1 and SPRED2 are increasingly understood, the physiological relevance of SPRED3 remains elusive. To elucidate its function, we generated SPRED3 knockout (KO) mice and performed phenotypic, molecular, and hormonal analyses. SPRED3-deficient mice exhibited growth retardation and a non-Mendelian genotype distribution. X-Gal staining revealed *Spred3* promoter activity in the thyroid, adrenal gland, pituitary, cerebral cortex, and kidney. Hormonal profiling identified elevated thyroid-stimulating hormone (TSH) and reduced thyroxine (T_4_) levels, indicating primary hypothyroidism. Thyroidal extracellular signal-regulated kinase (ERK) signaling was mildly reduced in SPRED3 KO mice, and immunoblotting revealed altered expression of autophagy regulators, including reduced sequestosome 1 (p62), increased autophagy-related gene 5 (ATG5), as well as an elevated microtubule-associated protein 1 light chain 3 (LC3) II/I ratio and a decreased pBeclin/Beclin ratio in SPRED3 KO mice. Our findings indicate that SPRED3 is involved in thyroidal homeostasis and plays a regulatory role in autophagy processes within the thyroid gland.

## 1. Introduction

SPRED proteins were first identified by Wakioka and colleagues in 2001 as negative regulators of the mitogen-activated protein kinase (MAPK) signaling pathway [1]. This pathway is activated by receptor tyrosine kinases (RTK) upon the binding of extracellular stimuli, such as epidermal growth factor (EGF), leading to a sequential activation of Ras, Raf, MEK, and ERK, with downstream effects on transcription factors like c-Myc and Elk-1 [2,3]. SPRED proteins, specifically SPRED1 and SPRED2, exert their inhibitory function by impeding the phosphorylation of Raf or by reducing Ras-GTP levels through their interaction with neurofibromin-1 (NF1), a GTPase-activating protein (GAP). Consequently, SPREDs are implicated in various biological processes, including cell motility, hematopoietic regulation, and tumor suppression [4,5].

While SPRED1 and SPRED2 have been well-characterized with regard to their roles in MAPK inhibition and physiological regulation, the biological functions of SPRED3 remain largely unexplored [6]. Although SPRED3 shares structural features with its homologs, including an N-terminal EVH1 domain and a C-terminal Sprouty-related (SPR) domain, it putatively lacks a functional c-Kit binding domain (KBD), which may account for its reduced inhibitory activity [7]. Human *Spred3* RNA expression is primarily restricted to the brain, glandular tissue, and mammary tissue, suggesting yet unknown tissue-specific roles distinct from those of SPRED1 and SPRED2 [6].

To date, elevated SPRED3 expression has been associated with higher survival rates in neuroblastoma patients and mutations in cervical carcinoma as well as glioblastoma [8,9]. Moreover, more recent findings link SPRED3 to thyroid carcinoma, where its overexpression correlates with poor clinical outcomes and advanced tumor stages [10,11]. Mechanistically, SPRED3 has been implicated in promoting thyroid cancer cell proliferation via the nuclear factor-κB (NF-κB) signaling pathway [10]. Furthermore, bioinformatic analyses suggest that SPRED3 may serve as a prognostic biomarker in thyroid carcinoma, possibly influencing extracellular matrix organization and immune cell infiltration [11]. The recent discovery of the first human mutation in SPRED3 further underscores its clinical significance [12].

Despite these findings, the precise molecular and physiological role of SPRED3 remains poorly understood. To address this gap, we generated a murine SPRED3 KO model to investigate its molecular and physiological impact. Functional validation included the verification of the KO model via RNA and protein expression analyses, alongside body weight measurements and Mendelian distribution assessments of offspring. X-Gal staining of brain, pituitary, kidney, thyroid, and adrenal gland revealed *Spred3* promoter activity that, in combination with the observed developmental malfunctions in SPRED3 KO mice, pointed to an endocrine role of SPRED3. Serum hormone levels of corticosterone, (nor-)adrenaline, dopamine, growth hormone (GH), TSH, and T_4_ were hence measured to evaluate endocrine function. Given indications of thyroid dysfunction in SPRED3-deficient mice, we hypothesized that impaired thyroglobulin processing, potentially involving macroautophagy regulators such as p62, microtubule-associated protein 1 light chain 3 (LC3), and ATG5, could contribute to the observed phenotype [13,14]. Western blot analyses of thyroid gland lysates from wild-type (WT) and KO mice were conducted to scientifically approach this hypothesis.

Our findings suggest that SPRED3 extends its role beyond oncogenic pathways, potentially regulating autophagic and lysosomal processes. This work provides a basis for understanding SPRED3’s function in thyroid biology. The established SPRED3 KO model offers a crucial platform for future research and lays the groundwork for deciphering the pathophysiological role of SPRED3 mutations in human disease [12].

## 2. Results

### 2.1. Generation and Validation of SPRED3 Knockout Mice

To investigate the role of SPRED3 in vivo, we generated SPRED3 KO mice using the embryonic stem cell (ESC) clone EPD0481_1 D11, derived from the C57BL/6N-A/a mouse line and provided by the International Knockout Mouse Consortium (Project 35886). The ESCs were modified to carry the tm1a (EUCOMM) Wtsi allele through homologous recombination, utilizing the vector PRPGS00122_A_D01 (Figure 1a). This strategy, known as gene targeting, permits precise genetic modifications, contrasting the random insertions seen in traditional gene trap methods [15,16].

The targeting vector incorporated a “KO-first” design, featuring a multifunctional cassette within the intron between exon 1 and exon 2 of the *Spred3* gene. This cassette comprises:A reporter trap element including:
○A splice acceptor (SA) sequence from the mouse *engrailed-2* gene.○A *lacZ* reporter gene for β-galactosidase expression.○A polyadenylation (Poly (A)) signal derived from Simian Virus 40, which truncates the endogenous transcription.
A *neomycin* resistance gene (neo) driven by a phosphoglycerate kinase (PGK) promoter, flanked by FRT sites for conditional removal.Three loxP sites enabling the potential for conditional gene deletion [17].

The genetically engineered ESCs were injected into isolated fertilized blastocysts of C57BL/6N mice and implanted into pseudo-pregnant females, after successful homologous recombination had been confirmed by X-Gal staining of the embryonic stem cells (Figure 1b). Resulting chimeric offspring were crossed with WT mice, producing heterozygous (Het) carriers of the modified allele. Homozygous SPRED3 KO mice were subsequently obtained by interbreeding heterozygous animals. To identify the genotypes of the mice relevant for this study, we used specific primers to amplify distinct DNA fragments via PCR (Figure 1c). Thus, we were able to verify the genotypes of the mice used in the study and proceed with our experimental setups involving the decipherment of the physiological impact of SPRED3.

To confirm successful knockout of murine SPRED3, we performed quantitative Reverse Transcription PCR (qRT-PCR) of SPRED3 WT and KO brain lysates, as well as Western blot analysis of adrenal gland lysates. qRT-PCR analysis revealed that GAPDH expression levels remained constant across WT and KO samples. However, *Spred3* transcript levels remained undetectable in KO mice (no amplification, Cq not reached), whereas WT mice exhibited clear amplification signals with averaged Cq values of 30 ± SEM (Figure 2a). This result confirms the effective transcriptional silencing of SPRED3 in KO animals.

Furthermore, Western blot analysis of adrenal gland lysates demonstrated the complete absence of SPRED3 protein in SPRED3 KO mice compared to WT controls, verifying the knockout at the protein level (Figure 2b).

Hence, we were able to assess the physiological impact of murine SPRED3 deficiency more thoroughly, as our data demonstrated the complete loss of SPRED3 expression both transcriptionally and translationally.

### 2.2. SPRED3 Deficiency Causes Growth Retardation and Deviations from Mendelian Ratios

Assessing the impact of SPRED3 deficiency on growth and survival, we monitored the body weight development of SPRED3 KO and WT mice over a period of up to 800 days. Nonlinear regression analysis revealed a distinctive reduction in body weight for both male and female SPRED3 KO mice compared to WT controls (Figure 3a,b).

We hence propose that SPRED3 plays a crucial role in normal physiological development and endocrine regulation [18].

To further scrutinize the developmental impact of SPRED3 deficiency on mice and investigate potential deviations from expected Mendelian inheritance, we genotyped offspring from heterozygous crosses. The analysis revealed a significant underrepresentation of homozygous SPRED3 KO mice compared to the expected Mendelian ratios (Figure 3c). While the number of WT and Het offspring closely matched expected distributions, the number of SPRED3 KO mice was significantly lower than predicted (*p* < 0.05, Chi-square test). Based on Mendelian expectations, 136 SPRED3 KO mice were anticipated, yet only 100 were observed. Hence, this significant deviation further underscores SPRED3’s potential involvement in developmental or postnatal viability processes and indicates that the complete loss of SPRED3 might impair survival.

### 2.3. SPRED3 Deficiency Does Not Alter Cardiac Performance

To evaluate whether SPRED3-deficient mice display additional systemic phenotypes, we further assessed hemodynamic parameters, given that SPRED2-deficient mice exhibit alterations in cardiac function [19]. However, SPRED3 KO and WT mice showed no differences in their hemodynamic profiles, suggesting that, unlike SPRED2, SPRED3 might not be critically involved in the regulation of cardiac performance under basal conditions (Table 1).

### 2.4. Scrutiny of Tissue-Specific Spred3 Promoter Activity

To gain an idea of which organs may be affected by SPRED3 deficiency and further assess the functionality of the knockout cassette, we performed X-Gal staining in KO mice. The generated SPRED3 KO model features a *lacZ* reporter gene that encodes β-gal (Figure 1a,b). The enzyme β-gal catalyzes the conversion of X-Gal into a blue product, staining the tissue and thereby demonstrating *Spred3* promoter-driven expression of the *lacZ* reporter gene in place of functional SPRED3 [20].

X-Gal staining revealed the most prominent blue staining, indicating high *Spred3* promoter activity, in the cerebral cortex, the hippocampal formation, and the amygdala (Figure 4a). The cerebral cortex is involved in coordinating and processing movements, language, and associations, whereas the hippocampus is crucial for spatial orientation, navigation, and memory formation [21,22,23]. The amygdala, however, located in the temporal lobe, is known for its involvement in processing emotions, particularly fear [24]. In contrast, subcortical regions exhibited only minimal *Spred3* promoter activity, suggesting a correlation between elevated expression and cell body density. Overall, we were able to replicate the findings of Kato et al. (2003) regarding SPRED3 expression in brain tissue. In contrast to their observations, however, we additionally identified *Spred3* promoter activity in the kidney, adrenal gland, pituitary, and thyroid gland [7]. The kidney displayed distinct activity, particularly in the cortex, where glomeruli and tubules are located [25]. In contrast, the medulla exhibited barely detectable staining. During kidney tissue preparation, a nerve was incidentally sectioned, which likely represents sympathetic fibers from the celiac plexus innervating the kidney (Figure 4b). This nerve also showed intense blue staining, indicating robust *Spred3* promoter activity (Figure 4b, arrow). In the adrenal gland, *Spred3* promoter activity was detected exclusively in the adrenal medulla, which originates from neural crest cells during ontogenesis and is responsible for the production of catecholamine hormones such as noradrenaline and adrenaline (Figure 4c) [26,27].

Similarly, the pituitary gland exhibited region-specific *Spred3* promoter activity: the posterior part (*neurohypophysis*) displayed weaker promoter activity, while the anterior part (*adenohypophysis*) showed moderately increased activity. The strongest activity, however, was observed in the intermediate part (*pars intermedia*) (Figure 4d).

Strikingly, *Spred3* promoter activity was also detected in thyroid follicles, indicating that SPRED3 might be potentially categorized as a hormonal regulator—especially taking into consideration that *Spred3* promoter activity was also observed in the pituitary and adrenal gland (Figure 4c–e).

Hence, our X-Gal stainings substantially indicate that SPRED3, like SPRED1 and SPRED2, is involved in specific endocytic or exocytic processes potentially governing hormonal circuits [28].

### 2.5. Hormonal Profiling Reveals Primary Hypothyroidism in SPRED3 KO Mice

To further assess our previous assumption of SPRED3 being involved in hormonal circuits, we analyzed serum hormone levels in SPRED3 WT and KO mice, focusing on hormones relevant to β-gal positive tissue regions (Figure 4a–e). These included adrenaline, noradrenaline, and dopamine for the adrenal medulla, GH and TSH for the pituitary gland, and T_4_ for the thyroid gland. To substantiate the validity of the X-Gal stainings, serum levels of corticosterone were also analyzed in SPRED3 WT and KO mice. As depicted in Figure 4c, *Spred3* promoter activity is exclusively observed in the adrenal medulla, and thus, it can be inferred that the molecular impact of SPRED3 is likewise restricted to this region of the adrenal gland. Based on this observation, serum levels of corticosterone, synthesized in the adrenal cortex, were expected not to differ significantly between SPRED3 WT and KO mice.

ELISA analyses of serum samples from SPRED3 WT and KO mice demonstrated that hormone concentrations of adrenaline, noradrenaline, dopamine, and corticosterone did not differ significantly between WT and KO samples (Figure 5a–d). In contrast, referring to corticosterone, SPRED2-deficient mice display distinctively elevated corticosterone levels, suggesting that SPRED2 and SPRED3 have organ-specific physiological functions [29].

Referring to GH, we found that GH levels were unaltered in mice younger than 120 days and significantly reduced in SPRED3-deficient mice older than 120 days (Figure 5e). This circumstance potentially suggests that SPRED3 might play a critical role in maintaining GH homeostasis as mice age. These findings are in line with the well-established neuroendocrine regulation of GH secretion, which is tightly controlled by the hypothalamic stimuli and feedback loops involving insulin-like growth factor 1 levels. The age-dependent decline observed in SPRED3-deficient mice may thus reflect disrupted regulatory feedback mechanisms [30].

Interestingly and more importantly, hormonal imbalances were not solely limited to GH. We also observed that SPRED3 KO mice exhibited significantly elevated TSH levels and reduced T_4_ concentrations, collectively indicating primary hypothyroidism (Figure 5f,g). In the thyroid gland, the relative colloid area was found to be significantly increased in SPRED3-deficient mice (Figure 5h,i), which was regarded as indicative of a stronger stimulation of the thyroid by the higher TSH levels, but this did not result in physiologically higher T4 levels. This warrants more comprehensive analyses of the molecular impact of SPRED3, particularly focusing on the thyroid gland, as key regulators of thyroid function and status are saliently altered in SPRED3-deficient mice and indicate SPRED3 is involved in the hormonal regulation of the hypothalamic–pituitary–thyroid axis.

### 2.6. ERK Signaling in the Thyroid Gland Is Reduced in SPRED3 KO Mice

Given that we identified hormonal alterations in SPRED3-deficient mice with a specific focus on the thyroid gland, we next aimed to investigate potential molecular signaling pathways involved in thyroid function. We focused on the expression levels of key components of the MAPK pathway, specifically ERK 1/2 and its phosphorylated (active) form (p-ERK 1/2), in the context of SPRED3 deficiency. This examination was prompted by the current lack of a comprehensive understanding of the molecular mechanisms underlying MAPK inhibition by SPRED3. To determine whether the observed thyroidal hormonal imbalance and the associated primary hypothyroidism are linked to MAPK pathway modulation dependent on functional SPRED3, we performed Western blot analysis with thyroid lysates from SPRED3 WT and KO mice. Hence, we were able to assess the expression levels of ERK 1/2 and p-ERK 1/2 (Figure 6a). Densitometric quantification revealed a slight but statistically significant reduction in the p-ERK/ERK ratio, with p-ERK 1/2 and ERK 1/2 levels being lower in KO samples compared to WT controls.

Additionally, both p-ERK 1/2 and total ERK 1/2 expression levels were significantly decreased in SPRED3 KO thyroid lysates relative to WT controls. Previous studies have shown that SPRED1 and SPRED2 negatively regulate the MAPK cascade by reducing phosphorylation, thereby impairing ERK 1/2 activation [4,5]. Based on this knowledge, an increase in p-ERK/ERK ratio in KO thyroid lysates was anticipated following SPRED3 loss. However, the opposite effect, a decrease in the p-ERK/ERK ratio, was observed (Figure 6c–e).

To explore whether SPRED1 might be upregulated as a compensatory response to SPRED3 deficiency, additional Western blot analysis was performed to assess its expression in thyroid lysates (Figure 6b). The corresponding densitometric quantification reveals a significant increase in SPRED1 protein level in KO samples compared to WT controls (Figure 6f). Furthermore, we asked if the formation of an SPRED3/SPRED1 heterodimer could be predicted by a computational approach using ChimeraX and AlphaFold4. Here, we found that SPRED3 and SPRED1 might also interact physically via their EVH1 and Sprouty domains. Especially the Sprouty-to-Sprouty interactions might be closer, as represented by the yellow lines, labelling a distance below 5 Ångström (Figure 6g). Taken together, these findings support the hypothesis that compensatory upregulation of SPRED1 in SPRED3-deficient mice may contribute to the reduced activation of the MAPK cascade. Further studies are hence necessary to thoroughly grasp the molecular mechanisms of SPRED3 in the thyroid gland within the context of MAPK signaling—a key signaling pathway in thyroid cancer [31].

### 2.7. SPRED3 Deficiency Alters Autophagy-Related Protein Expression in the Thyroid Gland

SPRED2 has been previously demonstrated to be involved in autophagic processes, interacting with LC3 via the LC3-interacting region (LIR) motif present in its SPR domain [32]. Considering that the LC3-binding residues are highly conserved between SPRED proteins (Figure 7a), we aimed to investigate whether, in addition to MAPK signaling, other pathways, such as macroautophagy, are dysregulated in the absence of SPRED3. Our analysis focused on the expression profiles of core autophagy components: Beclin, LC3, p62, and ATG5. Phosphorylation of Beclin by the kinases AMPK or ULK1 is a critical step in the initiation of autophagy. LC3 conversion serves as a general read-out marker for autophagic flux, while p62 functions as a selective autophagy receptor, and ATG5 is crucial for autophagosome formation [33]. We hypothesized that the reduced T_4_ serum levels observed in SPRED3 KO mice might be attributed to dysregulated thyroglobulin processing in thyrocytes, which shares molecular similarities with macroautophagy [13]. Macroautophagy, commonly referred to as autophagy, is a highly regulated self-digestion and recycling process crucial for cellular homeostasis. It specifically involves the sequestration of cellular components within double-membraned vesicles (autophagosomes), which subsequently fuse with lysosomes for degradation and recycling of macromolecules [34].

To preliminarily screen how and if macroautophagic functionality is impaired due to SPRED3 deficiency, we performed Western blot analyses to examine the relative levels of p-Beclin/Beclin ratios, LC3-II/I ratios, and expression levels of p62 and ATG5 in thyroid gland lysates of SPRED3 WT and KO mice (Figure 7b and Figure 8e).

Densitometric analyses of the ratio between cleaved active LC3-II to the inactive LC3-I revealed an increased ratio of LC3-II/I in SPRED3-deficient mice as compared to WT controls (Figure 7c). This can be regarded as an increase in autophagic flux; however, LC3-II is degraded during autophagy, as well, therefore these results might also reflect a disturbance in late processes of autophagy, regulated by SPRED3. As described for other SPRED proteins, SPRED3 might also interact directly with LC3 via its LIR domain. To address this question, we investigated a possible interaction again by a computational approach using ChimeraX and AlphaFold4. In this case, the interaction of LC3 and SPRED3 seems to be mediated by the Sprouty domain of SPRED3, containing the LIR motif (Figure 7d).

To further clarify disturbances in autophagic processes, we tested whether the autophagy adaptor p62, which also interacts with other SPRED proteins, might also interact with SPRED3 and if expression levels of p62 might be altered in SPRED3-deficient mice. Computational prediction of the interaction suggested an interaction of p62 with the Sprouty domain of SPRED3 (Figure 8a), which was supported by a positive co-immunoprecipitation of p62 by using antibodies for SPRED3 (Figure 8c). Similarly, the interaction prediction of ATG5 with SPRED3 was positive, interestingly, in this case mediated by the EVH1 domain of SPRED3 (Figure 8b), and these data were again supported by a positive co-immunoprecipitation result (Figure 8d).

Quantification of expression levels of these key regulators of autophagy revealed a decrease in the ratio of pBeclin to Beclin (Figure 8f), a decrease in p62 levels (Figure 8g), and an increase in ATG5 levels (Figure 8h) in SPRED3-deficient mice.

To sum up, our results provide support to the hypothesis that key autophagic regulators and SPRED3 interact and that the loss of SPRED3 disturbs the homeostasis of autophagy in the thyroid gland.

## 3. Discussion

The molecular role of SPRED3 has so far been unraveled primarily in vitro, significantly limiting the experimental scope for deciphering its physiological function [7,10,11]. In contrast, our study provides the first comprehensive analysis of SPRED3 deficiency in an entire intact model organism, allowing us to decode the molecular role of SPRED3 in vivo. We validated the functionality of our SPRED3 KO mouse line at both the transcriptional and translational levels, enabling us to investigate the physiological implications of SPRED3 deficiency across various tissues. Using this KO model, we demonstrated that SPRED3-deficient mice not only exhibit reduced body weight but also develop primary hypothyroidism, as evidenced by significantly elevated TSH levels and decreased T_4_ serum concentrations. This endocrine dysregulation aligns with the tissue-specific SPRED3 promoter activity identified via X-Gal staining, which provided the first indication that SPRED3 may play a critical role in thyroid function. Although recent in vitro studies have suggested that SPRED3 could serve as a potential biomarker in thyroid carcinoma, the physiological role and molecular mechanism of SPRED3 in this context have yet to be fully elucidated [10,11].

Through further Western blot analyses focusing on the thyroid gland, we evidently illustrated that SPRED3 deficiency leads to compensatory regulation within the MAPK signaling pathway. More specifically, our findings indicate that SPRED3 might interact with unique and yet unknown upstream effectors or modulate alternative feedback loops, potentially leading to context-specific effects [6,35,36]. In addition, compensatory mechanisms involving other SPRED isoforms, such as upregulation of SPRED1 or regulatory proteins such as dual-specificity phosphatases (DUSP), might be activated in the absence of SPRED3, thereby dampening the MAPK cascade despite the loss of one inhibitor [37].

Moreover, we postulated that, analogous to SPRED2, SPRED3 might influence the expression profile of autophagy-related regulators [32]. The study by Jiang et al. [32] focused on the impact of SPRED2 on LC3-I/-II expression, which we aimed to investigate in SPRED3-deficient thyroid tissue. Our findings revealed an increased ratio of LC3-II/I in SPRED3 KO mice. This can be seen as an increased autophagic flux. And as LC3-II is degraded during autophagy, these results might also reflect a disturbance in late processes of autophagy, regulated by SPRED3.

Phosphorylation of Beclin by different kinases is a critical step in the initiation of autophagy. Quantification of expression levels of this key regulator of autophagy revealed a decrease in the ratio of pBeclin to Beclin.

The observed significant upregulation of ATG5, a crucial factor in autophagosome formation, could indicate enhanced initiation of the autophagic process in the absence of SPRED3. ATG5 is a component of the ATG12-ATG5-ATG16L complex, which is essential for membrane elongation and autophagosome maturation [33]. Simultaneously, the significant reduction in p62 levels in KO samples could initially suggest efficient autophagic degradation, as p62 is typically degraded during selective autophagy when cargo is successfully delivered to lysosomes [38]. However, such an interpretation remains ambiguous, as effective cargo degradation relies not only on autophagosome formation but also on intact transport and fusion mechanisms—both of which may be compromised in the absence of SPRED3 [39,40]. An impairment in autophagosome–lysosome fusion in SPRED3-deficient thyroid tissue is, therefore, conceivable. In epithelial cells such as thyrocytes, lysosomes are predominantly localized in the perinuclear region, necessitating efficient long-range transport of autophagosomes along microtubules [39].

A disturbance in this transport process may impede fusion and result in the disturbance of the degradation of cargo, including p62 and LC3-II. This potential interference in autophagosome trafficking and degradation provides a mechanistic link to the initial hypothesis that SPRED3 may contribute to the autophagy-like processing of thyroglobulin. Since thyroglobulin undergoes intracellular degradation in endolysosomes following endocytosis or macroautophagy-like uptake, a functional role of SPRED3 in regulating vesicle transport or lysosomal processing could explain both the altered expression of autophagy-related proteins and the molecular mechanisms of the observed hypothyroidism in SPRED3 KO mice [13].

To further substantiate our hypothesis of SPRED3 being involved in this mechanism with regard to its interaction with crucial autophagy-related proteins, we utilized ChimeraX software alongside AlphaFold structure predictions to evaluate the interaction potential between SPRED3 and key components of autophagy (LC3, p62, and ATG5). Our in-silico analyses revealed that the molecular interaction between these proteins and SPRED3 is indeed plausible. SPRED3 might interact directly with p62 and ATG5, as described for other SPRED proteins. These data were supported by positive co-immunoprecipitation results. Hence, further studies like co-localization studies are indispensable to gain deeper insights into the physiological impact of SPRED3 [41].

Collectively, the data indicate that SPRED3 deficiency might interfere with later stages of the autophagic process—particularly autophagosome transport or fusion. Definitive conclusions regarding autophagic flux, so far not definitely determined, require follow-up experiments with additional markers of lysosomal degradation, such as lysosomal-associated membrane proteins 1 and 2. Fluorescence-based co-localization approaches or treatment of isolated cells with lysosomal inhibitors such as bafilomycin A1 may provide more detailed insights into autophagosome maturation and turnover and could help elucidate the molecular mechanisms underlying the murine hypothyroidism associated with SPRED3 deficiency [38].

Further studies using the SPRED3 KO mouse line that aim to elucidate the function of SPRED3 within the thyroid gland could benefit from proteomics or phospho-proteomics analyses. Such investigations would allow for a broad-spectrum evaluation of whether and how SPRED3 influences lysosomal processes within the thyroid and whether the deviations in offspring numbers are indeed attributable to developmental disturbances and dysregulations in energy metabolism.

In sum, such investigations may not only clarify the role of SPRED3 in thyroid physiology but also contribute to understanding its hypothetical involvement in human endocrine and metabolic disorders, especially considering that a four-year-old girl carrying a heterozygous SPRED3 mutation has already been identified [12].

## 4. Materials and Methods

### 4.1. SPRED3 Mutant Mice

SPRED3 KO mice were generated by blastocyst injection of the embryonic stem cell line D11 (International Knockout Mouse Consortium, Project 35886). The embryonic cells were injected into blastocysts and re-implanted into the uterus of pseudo-pregnant mice. Crossing of offspring led to homozygous SPRED3 KO mice. WT mice were used as controls. To minimize possible inbred effects in this study, mice were raised on a mixed C57Bl/6 × 129/Ola genetic background. Mice were kept in plastic cages on a daily 12 h light/dark cycle under controlled room temperature (21 ± 1 °C), humidity of 55 ± 5%, and tap water and standard mouse chow ad libitum unless stated otherwise. All experiments were conducted with organs from adult mice at the age of 8 to 12 months. All experiments were approved by the local councils for animal care and were conducted according to the guidelines of the European Union (2010/63/EU).

### 4.2. X-Gal Staining

X-Gal stainings of murine SPRED3 KO tissue or cells were performed as described previously [18,42]. Tissue sections were captured using a REX lighting plate (Carl Roth; Karlsruhe, Germany) and a digital camera (Sony, A6400; San Mateo, CA, USA).

### 4.3. Genotyping of Mice

Offspring were genotyped using the following set of primers: for WT PCR, amplifying a 772 bp fragment, forward primer 1 (5′-CGCAGGAATCCAGCTGTGAGAG-3′) and reverse primer 2 (5′-AAACAACACAGGCCGTGAACTCAAA-3′; binding to the gene targeting cassette), for knockout PCR, amplifying a 600 bp fragment, primer 1 and reverse primer 3 (5′-CCCAGGCTTCACTGAGTCTCTGGCATCT-3′; binding to the WT allele). PCR was performed with an annealing temperature of 60 °C and 30 amplification cycles.

### 4.4. RNA Extraction and RT-qPCR Analyses

Total RNA was extracted from mouse brain tissue of SPRED3 KO and WT controls using the Total RNA Isolation Reagent (AB-0303, ABgene; Altrincham, England) according to the manufacturer’s instructions. Briefly, brain tissue was homogenized (Polytron PT 3100, Kinematica; Malters, Switzerland), followed by phase separation with chloroform and precipitation with isopropanol. The RNA pellet was washed with ethanol, dried, and resuspended in 50 µL of 0.1% diethylpyrocarbonate-treated, RNAase-free water. RNA concentration was measured with the SimpliNano spectrophotometer (15205928, biochrom; Berlin, Germany).

For reverse transcription, 1 µg of total RNA was transcribed into cDNA using the Transcriptor First Strand cDNA Synthesis Kit (04896866001, Roche; Basel, Switzerland) with anchored-oligo (dT)18 primers following the manufacturer’s protocol.

Quantitative PCR was performed using TaqMan Gene Expression Assays for SPRED3 and GAPDH as a control (4331182, Applied Biosystems; Foster City, CA, USA). All steps were conducted according to the manufacturer’s instructions. qPCR was performed on the LightCycler 96 System (05815916001, Roche) with an annealing temperature of 60 °C and 45 amplification cycles.

### 4.5. Western Blotting and Densitometric Quantification of Detected Bands

Thyroid and adrenal gland lysates were prepared in homogenization buffer (150 mM NaCl, 1% IGEPAL, 0.5% sodium deoxycholate, 50 mM Tris, pH 8) supplemented with cOmplete Protease Inhibitor and PhosSTOP Phosphatase Inhibitor (04693116001 and 04906837001, Roche, Basel, Switzerland). Protein concentrations were estimated utilizing the Qubit 4 Fluorometer (Q33239, Thermo Scientific; Waltham, MA, USA) with the Protein Broad Range Assay Kit (A50668, Thermo Scientific) following the manufacturer’s protocol. Lysates were diluted 1:6 with 6× SDS gel loading dye (1 M Tris-HCl, pH 6.8, 10% SDS, 0.2% bromophenol blue, 70% glycerol), separated by SDS-PAGE on 10% gels and transferred to nitrocellulose membranes (Amersham Protran, 10600001, Cytiva; Marlborough, MA, USA) using a semi-dry blotting system (custom-built).

For Western blot analyses, the following antibodies were used: affinity-purified polyclonal rabbit anti-SPRED3 (Antigen AS 156–287; generated by Eurogentech, Seraing, Belgium) according to the manufacturer’s protocol [28], polyclonal rabbit anti-phospo-ERK (9101, Cell Signaling Technology; Danvers, MA, USA), polyclonal rabbit anti-ERK (9102, Cell Signaling Technology; Danvers, MA, USA), polyclonal rabbit anti-SPRED1 (AS 124–233 [28]), monoclonal rabbit anti-Phospho-Beclin-1 (S93) (D9A5G) (14717, Cell Signaling Technology; Danvers, MA, USA), monoclonal rabbit anti-Beclin-1 (D40C5) (3495, Cell Signaling Technology; Danvers, MA, USA), monoclonal rabbit anti-LC3A/B (12741, Cell Signaling Technology; Danvers, MA, USA), monoclonal rabbit anti-p62 (7695S, Cell Signaling Technology; Danvers, MA, USA), monoclonal rabbit anti-ATG5 (12994, Cell Signaling Technology; Danvers, MA, USA), monoclonal rabbit anti-GAPDH (2118S, Cell Signaling Technology; Danvers, MA, USA), and peroxidase-conjugated goat anti-rabbit IgG (111035144, Jackson ImmunoResearch; West Grove, PA, USA)—all diluted in 5% non-fat dry milk (A0830, AppliChem; Darmstadt, Germany) in PBS (137 mM NaCl, 2.7 mM KCl, 10 mM Na_2_HPO_4_, 2 mM KH_2_PO_4_, pH 7.4) supplemented with 0.05% Tween 20 (A1389, AppliChem). For signal detection, the chemiluminescent substrate (WesternBright Sirius, 541019, Biozym; Wien, Vienna) was prepared and applied to the membranes. The signal was captured using image capture software VWR Imager CHEMI Premium 1.8.2.0 (Avantor Sciences, Radnor, PA, USA). After conversion to TIFF files, band pixel densities were quantified using ImageJ software (National Institutes of Health, version 1.54d).

### 4.6. Examination of Serum Hormone Levels

To obtain murine serum examining the hormonal status of SPRED3 KO mice, SPRED3 WT and KO mice were anesthetized via CO_2_ inhalation, and subsequently decapitated. Trunk blood was collected immediately after decapitation into Eppendorf tubes and allowed to clot at room temperature for 40 min to 1 h. After centrifugation at 10,000× *g* for 3 min, the serum was retrieved and utilized for commercially available ELISA kits: adrenaline, noradrenaline, dopamine (catecholamine kit, KA1880, Abnova; Taipei City, Taiwan), corticosterone (RTC002R, Bio Vendor; Brno, Czech Republic), GH (orb553478, Biorbyt; Cambridge, UK), TSH (orb409000, Biorbyt), and T_4_ (orb1173295, Biorbyt). All serum collections were performed in the morning to minimize circadian variation of hormone levels. Hormonal concentrations were calculated based on the standard curve using a four-parameter logistic regression model (Microplate reader Infinite M200 Pro, TECAN; Software Magellan, version Pro 7.4; Männedorf, Switzerland).

### 4.7. Invasive Hemodynamics

Measurements of left ventricular pressure/volume loops were performed under 1.5% isoflurane anesthesia with a 1.4 F PV catheter (SPR-839NR) and an MPVS Ultra pressure-volume loop system (Millar Instruments; Houston, TX, USA). Data were collected for 30 sec with a PowerLab 8/35 and analyzed with the P/V loop module of LabChart Pro 8.1.30 (ADInstruments; Sydney, Australia). Animals with a heart rate lower than 420/min were excluded from the analysis.

### 4.8. Co-Immunoprecipitation Assays

For co-immunoprecipitation assays, brain cortices were lysed in 1.5 mL ice-cold homogenization buffer (see above) and cleared by centrifugation (17,000× *g*, 4 °C, 10 min). The supernatants were precleared with 50 µL ROTIGarose Protein A/G beads (Carl Roth, Karlsruhe, Germany) for 1 h at 4 °C, and subsequently the resulting supernatants were incubated with 20 µL polyclonal rabbit anti SPRED3 antibodies (see above) for 1 h at 4 °C. To precipitate the immune complexes, 50 µL ROTIGarose Protein A/G beads were added to each sample, and binding was allowed for 1 h at 4 °C. Beads with bound immune complexes were recovered and washed three times by centrifugation (11,000× *g*, 4 °C, 3 min). Bound proteins were eluted by the addition of 50 µL SDS sample buffer and analyzed by Western blotting (see above). To detect p62 and ATG5, we used polyclonal goat anti-p62/SQSTM1 (dilution 1:200; sc-10117, Santa Cruz, Dallas, USA) and polyclonal goat anti-ATG5 (dilution 1:200; sc-8666, Santa Cruz, Dallas, TX, USA), respectively.

### 4.9. ChimeraX

To investigate potential interaction partners of SPRED3, molecular modeling and structural analysis were performed using ChimeraX (Version 1.8rc202405310000) and AlphaFold4. ChimeraX is a powerful tool for visualizing and analyzing molecular structures, enabling detailed three-dimensional representations of protein structures and the assessment of protein–protein interactions [43,44]. AlphaFold4, a state-of-the-art software for protein structure prediction, was utilized to generate theoretical models of SPRED3 interactions with other proteins. AlphaFold4 leverages advanced algorithms and machine learning techniques to predict three-dimensional conformation of proteins based on their amino acid sequences. The predicted protein structures were subsequently imported into ChimeraX for further analysis, enabling validation of the interactions.

### 4.10. Statistical Analysis

Statistical analyses were performed using GraphPad Prism 10 (GraphPad Software, La Jolla, CA, USA). If data sets were compared, the mean value of WT samples was set to 1 to allow for the comparison of relative expression changes, and the mean value of KO samples was expressed as *x*-fold of WT. The normal distribution was tested using the Shapiro–Wilk test. If a normal distribution was present, unpaired and two-tailed Student’s *t*-tests were performed. If there was no normal distribution, the Mann–Whitney U test was utilized. Results depicted as scatter dot plots are expressed as mean ± standard error of mean (SEM). Assessing body weight development, nonlinear regression (curve fit) was employed, and to evaluate Mendelian distribution, we applied Chi-square test. A *p*-value of 0.05 or smaller was considered statistically significant.

## Figures and Tables

**Figure 1 ijms-26-07660-f001:**
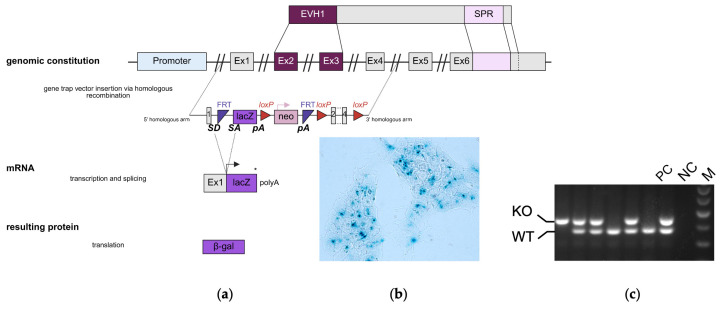
Generation of the SPRED3 KO mouse line, as well as genotyping of SPRED3 WT, Het, or KO mice by PCR analysis. (**a**) The gene trap vector (PRPGS00122_A_D01) includes elements like the reporter gene *lacZ*, a *neomycin* (neo) resistance gene, FRT sites, as well as loxP sites. The flanking upstream splice acceptor and downstream transcriptional termination sequences (polyadenylation sites) ensure that transcription is terminated and that the target gene is no longer transcribed. Insertion of this vector was conducted via homologous recombination, enabling precise genetic modification. Ultimately, the resulting protein consists solely of β-galactosidase (β-gal). For our purposes, it was not mandatory to make use of the conditional potential of the KO construct. Created in BioRender; (**b**) Representative X-Gal staining of cultured EPD0481_1 D11 embryonal stem cells demonstrating functionality of the SA, necessary for efficient SPRED3 knockout; (**c**) Genotyping of mice via PCR. The PCR products were separated by agarose gel electrophoresis to distinguish between WT, Het, or KO genotypes. WT mice displayed a single band at 368 bp, whereas KO mice showed a single band at 379 bp. Het mice exhibited both bands, confirming the presence of a WT and a KO allele. FRT = Flippase recognition target, loxP = locus of X-over P1, SD = splice donor, SA = splice acceptor, pA = polyadenylation site, Ex = exon. PC = positive control (heterozygous mouse), NC = negative control (H_2_O), M = marker.

**Figure 2 ijms-26-07660-f002:**
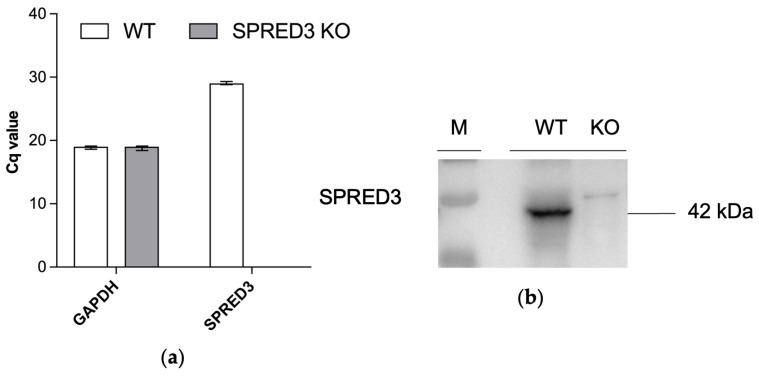
Transcriptional and translational verification of SPRED3 knockout mouse line. (**a**) Comparative Cq values for GAPDH and SPRED3 in WT and KO samples after 45 cycles. The Cq value is plotted on the *y*-axis, where a lower value indicates a higher initial amount of target cDNA. While GAPDH (left) showed nearly identical Cq values in both WT and KO samples, SPRED3 (right) was exclusively detected in WT mice. Data are presented as mean ± SEM (WT and KO *n* = 5); (**b**) Expression of SPRED3 in adrenal gland lysates of SPRED3 KO mice. Representative Western blot showing SPRED3 expression in adrenal gland lysates of SPRED3 D11 KO mice in comparison to WT control.

**Figure 3 ijms-26-07660-f003:**
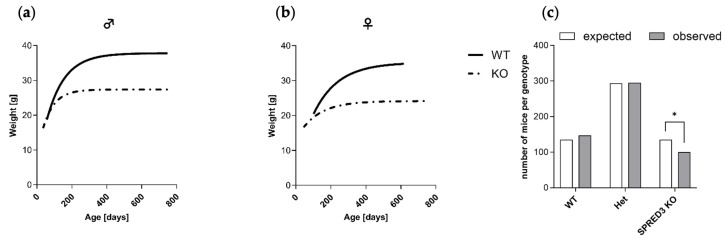
Impact of SPRED3 deficiency on growth dynamics and genotypic distribution. (**a**,**b**) Nonlinear regression curves representing body weight development over 800 days in male (left) and female (right) WT and SPRED3 KO mice (WT male *n* = 28, KO male *n* = 37, WT female *n* = 24, KO female *n* = 35). SPRED3 KO males and females exhibit distinctively reduced body weight compared to WT littermates throughout development. Data are represented as weight in grams [g] plotted against age in days; (**c**) Genotypic distribution of offspring from heterozygous crosses, showing the expected (white bars) versus observed (filled bars) numbers of WT, Het, and SPRED3 KO mice (WT expected *n* = 135, Het expected *n* = 294, KO expected *n* = 135, WT observed *n* = 147, Het observed *n* = 295, KO observed *n* = 100). A significant reduction in SPRED3 KO mice was observed compared to Mendelian expectations. Data are presented as absolute counts of mice per genotype. * *p* < 0.05.

**Figure 4 ijms-26-07660-f004:**
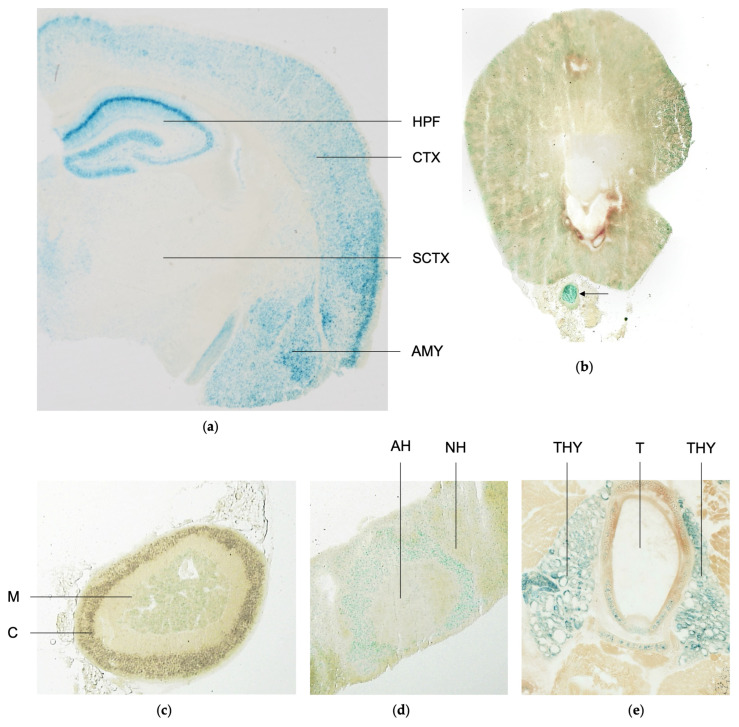
X-Gal staining of SPRED3 KO mouse tissue indicates *Spred3* promoter activity in the brain, kidney, adrenal gland, pituitary, and thyroid gland. (**a**) Coronal brain sections depict strong staining within the hippocampal formation, cerebral cortex, and amygdala; (**b**) In the kidney, prominent promoter activity was predominantly observed in the cortical region. Additionally, intense staining was detected in a nerve section (arrow); (**c**) In the adrenal gland, promoter activity was exclusively detected in the adrenal medulla; all other regions of the adrenal gland showed no detectable *Spred3* promoter activity; (**d**) The pituitary gland exhibited the strongest staining in the *pars intermedia*; (**e**) In the thyroid gland, high *Spred3* promoter activity was observed in the thyroid follicles. CTX = cerebral cortex, HPF = hippocampal formation, AMY = amygdala, SCTX = subcortical areas, M = medulla, C = cortex, AH = adenohypophysis, NH = neurohypophysis, THY = thyroid gland, T = trachea.

**Figure 5 ijms-26-07660-f005:**
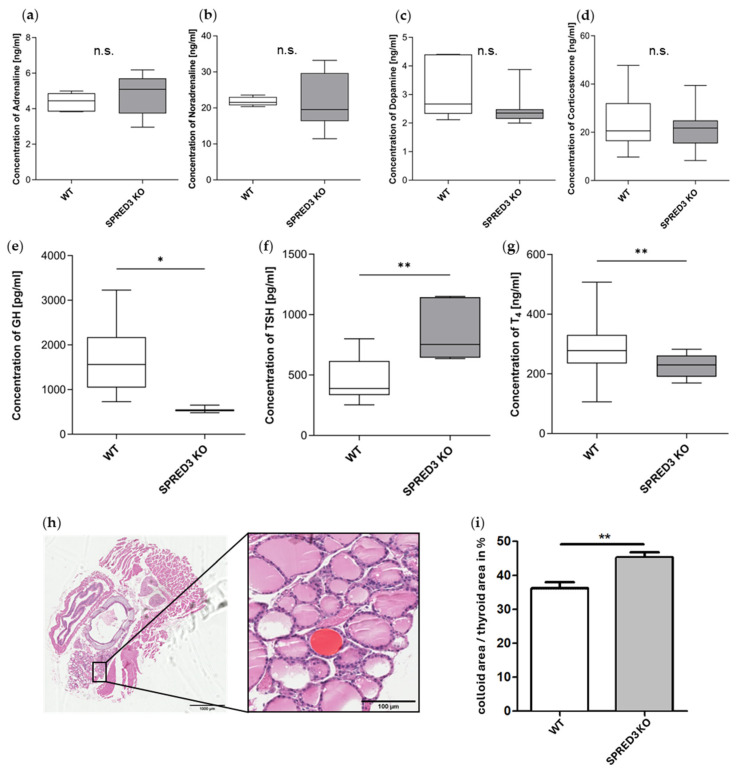
Hormonal analyses reveal primary hypothyroidism in SPRED3-deficient mice, characterized by elevated TSH and reduced T_4_ levels. (**a**–**d**) Serum concentrations of adrenaline (WT *n* = 6, KO *n* = 9), noradrenaline (WT *n* = 5, KO *n* = 9), dopamine (WT *n* = 6, KO *n* = 9) and corticosterone (WT *n* = 12, KO *n* = 17) remain unchanged between SPRED3 WT and KO samples; (**e**) GH serum levels of SPRED3 WT and KO mice older than 120 days indicate significant decrease within KO samples (WT *n* = 8, KO *n* = 3); (**f**,**g**) Hormonal analyses of the thyroid axis reveal primary hypothyroidism in SPRED3 KO mice with significantly increased TSH (WT *n* = 8, KO *n* = 5) and markedly reduced T_4_ levels (WT *n* = 21, KO *n* = 22). (**h**) Exemplary hematoxylin/eosin-stained [5] tissue section comprising thyroid gland (left), and detail enlargement (right) as used for the estimation of the ratio of colloid area (red) to thyroid area. (**i**) Increased relative colloid area in SPRED3-deficient mice (WT *n* = 11 sections, KO *n* = 8 sections). * *p* < 0.05, ** *p* < 0.01. n.s. = not significant.

**Figure 6 ijms-26-07660-f006:**
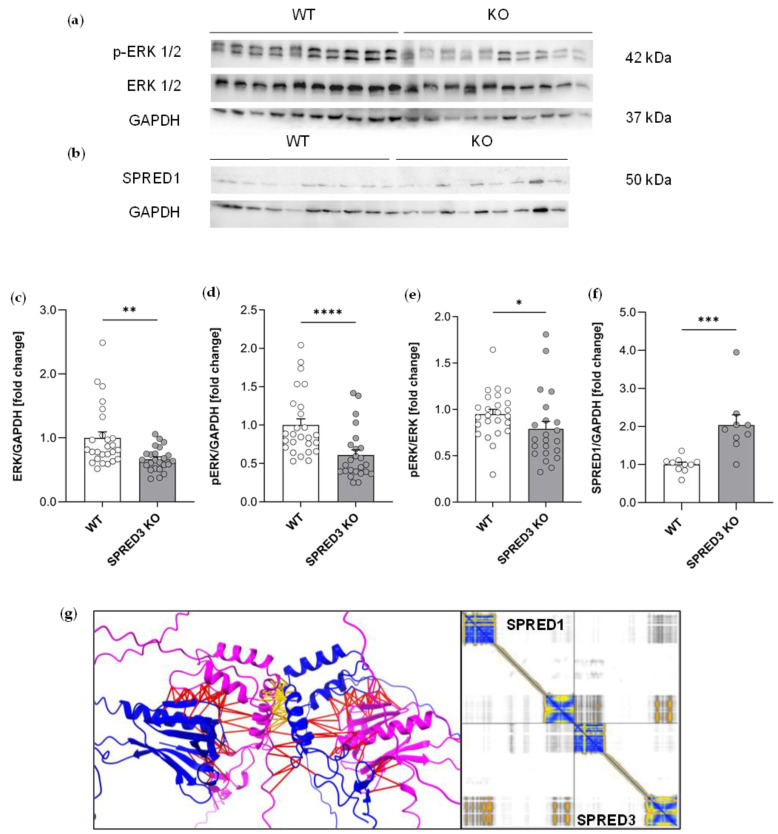
SPRED1 compensates for SPRED3 deficiency in mice, causing a significantly reduced p-ERK/ERK ratio. (**a**,**b**) Representative Western blots illustrating the expression of p-ERK1/2, ERK1/2 (WT/KO *n* = 10), and SPRED1 (WT *n* = 10, KO *n* = 9) in thyroid gland lysates from SPRED3 KO mice and WT controls. GAPDH was used as a loading control to ensure equal protein loading; (**c**–**f**) Densitometric quantification of protein levels normalized to GAPDH. The phosphorylation ratio of p-ERK1/2 to total ERK1/2 was also calculated. The quantification indicates a significant decrease in the p-ERK 1/2 to ERK 1/2 ratio in SPRED3 KO mice compared to WT controls. (**g**) Best model of structure and interaction prediction by ChimeraX. SPRED3 blue, SPRED1 magenta, distance below 5 Ångström in yellow, distance above 5 Ångström in red. The error plot on the right side shows an exact prediction of the EVH1 and Sprouty domains (yellow and blue colors) as well as a possible interaction between the EVH1 and Sprouty domains (yellow color). Data are presented as mean ± SEM. Dots represent the number of used mice (WT p-ERK 1/2 and ERK 1/2 *n* = 26, KO p-ERK 1/2 and ERK 1/2 *n* = 24; WT SPRED1 *n* = 10, KO SPRED1 *n* = 9). * *p* < 0.05, ** *p* < 0.01, *** *p* < 0.001. **** *p* < 0.0001.

**Figure 7 ijms-26-07660-f007:**
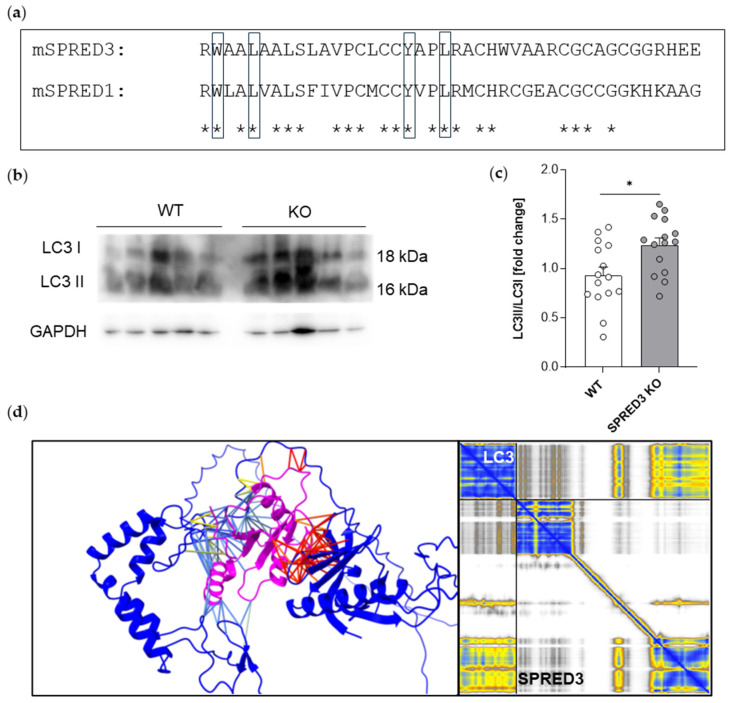
SPRED3 KO mice exhibit altered expression profiles of key autophagic regulator LC3. (**a**) Sequence alignment of murine SPRED1 and SPRED3 (mSPRED3; mSPRED1) within the SPR domain containing the LIR motifs (rectangles) highlight salient conservation of amino acids essential for efficient binding of LC3; (**b**) Representative Western blots illustrate the expression of LC3 (WT *n* = 5, KO *n* = 5), in thyroid gland lysates from SPRED3 D11 KO mice and WT controls. GAPDH was used as a loading control to ensure equal protein loading; (**c**) Densitometric quantification of protein levels. Quantification analysis shows an increased LC3-II/I ratio in SPRED3 KO thyroid lysates relative to WT controls (WT *n* = 15, KO *n* = 15). (**d**) Best model of structure and interaction prediction by ChimeraX. SPRED3 blue, LC3 magenta, distance below 5 Ångström in yellow, distance above 5 Ångström in red. The error plot on the right side shows an exact prediction of the EVH1 and Sprouty domain of SPRED3 and an exact prediction of LC3 structure (yellow and blue colors), as well as a possible interaction between the Sprouty domain, containing the LIR and LC3 (yellow color). Data are presented as mean ± SEM. Dots represent the number of used mice. * *p* < 0.05.

**Figure 8 ijms-26-07660-f008:**
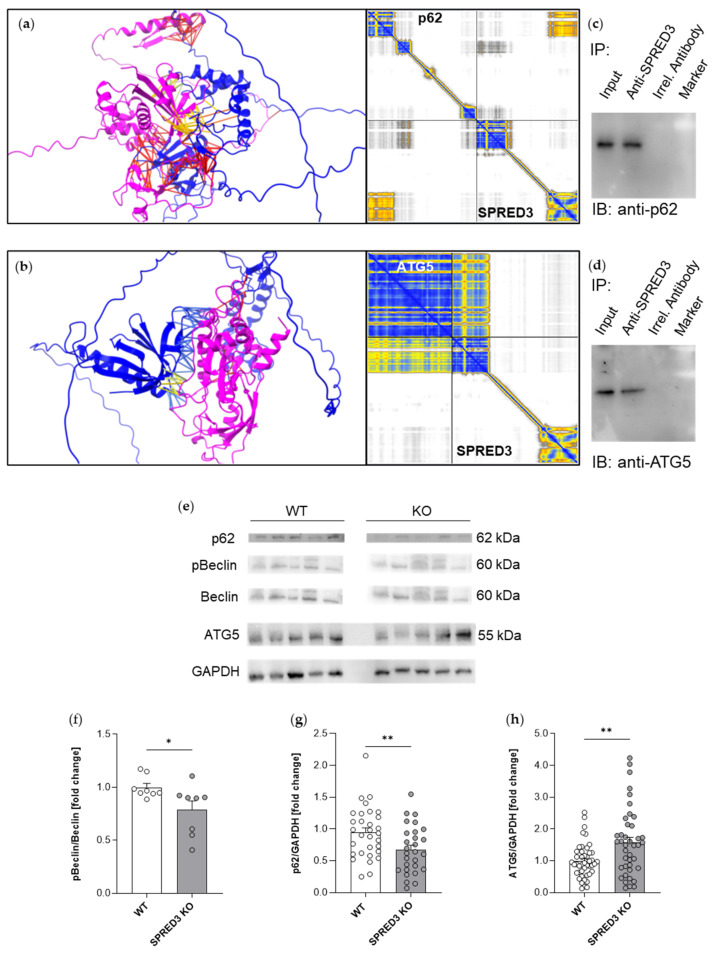
SPRED3 KO mice exhibit altered expression profiles of key autophagic regulators involving p62, Beclin, and ATG5. (**a**,**b**) Best models of structure and interaction predictions by ChimeraX. SPRED3 blue, p62 magenta (**a**), SPRED3 blue, ATG5 magenta (**b**), distance below 5 Ångström in yellow, distance above 5 Ångström in red. The error plots on the right side shows an exact prediction of the EVH1 and Sprouty domain of SPRED3 and an exact prediction of the N-terminal domain of p62 and ATG5 structure (yellow and blue colors) as well as a possible interaction between the Sprouty domain and p62, and the EVH1 domain with ATG5 (yellow and blue); (**c**,**d**) Co-immunoprecipitation with anti-SPRED3 co-precipitated p62 (**c**) and ATG5 (**d**). IP: precipitating antibody, irrel.: irrelevant, IB: antibodies used in subsequent immunoblot. (**e**) Representative Western blots illustrate the expression of p62, pBeclin, Beclin, and ATG5 in thyroid gland lysates from SPRED3 KO mice and WT controls. GAPDH was used as a loading control to ensure equal protein loading. (**f**–**h**) Densitometric quantification of protein levels. Quantification analysis revealed an increase in the pBeclin/Beclin ratio and a significant decrease in p62 levels in SPRED3 KO thyroid lysates relative to WT controls. ATG5 was significantly upregulated in the KO group. Data are presented as mean ± SEM. Dots represent the number of used mice (WT pBeclin/Beclin *n* = 8, KO *n* = 8; WT p62 *n* = 31, KO p62 *n* = 29; WT ATG5 *n* = 45, KO ATG5 *n* = 41). * *p* < 0.05, ** *p* < 0.01.

**Table 1 ijms-26-07660-t001:** Hemodynamic parameters of WT and SPRED3-deficient mice.

Hemodynamic Parameters (Isoflurane Anesthesia, ±SEM)	WT (*n* = 13)	SPRED3-KO (*n* = 12)
HR (min^−1^)	488 ± 12.5	497 ± 17.8
SV (µL)	15.3 ± 0.6	15.5 ± 0.7
ESV (µL)	17.4 ± 1.2	16.5 ± 1.5
EDV (µL)	30.8 ± 1.6	30.5 ± 1.3
EF (%)	49.6 ± 2.1	51.9 ± 2.9
CO (µL min^−1^)	7426.6 ± 212.9	7807.6 ± 542.2
ESP (mmHg)	95.6 ± 2.7	94.8 ± 3.5
EDP (mmHg)	8.5 ± 1.3	6.4 ± 0.7
SW (mmHg × µL)	1208.0 ± 63.6	1255.3 ± 70.8
V@dP/dt max (µL)	31.3 ± 1.4	30.4 ± 1.2
V@dP/dt min (µL)	16.9 ± 1.2	16.0 ± 1.3
Ea (mmHg µL^−1^)	6.3 ± 0.3	6.2 ± 0.4
**Systolic indices**		
dP/dt max (mmHg s^−1^)	7657.1 ± 263.5	8465.8 ± 682.4
dV/dt max (µL s^−1^)	642.9 ± 25.3	693.2 ± 37.8
**Diastolic indices**		
-dP/dt min (-mmHg s^−1^)	8385.2 ± 426.8	9284.7 ± 637.0
Tau (W) (ms)	6.7 ± 0.4	5.9 ± 0.4

HR: heart rate; SV: stroke volume; ESV: end-systolic volume; EDV: end-diastolic volume; EF: ejection fravtion; CO: cardiac output; ESP: end-systolic pressure; EDP: end-diastolic pressure; SW: stroke work; V@dP/dt max: volume at dP/dt max; V@dP/dt min: volume at dP/dt min; Ea: arterial elastance; dP/dt max: maximal rate of pressure increase; dV/dt max: maximum dV/dt/; dP/dt min: maximal rate of pressure decline; Tau (W): Relaxation constant according to Weiss method. *n* = 13 mice of each genotype, no parameters were found to be significantly altered.

## Data Availability

Data are available on request from the corresponding authors.

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
