# Peer review of "Loss of SPRED3 Causes Primary Hypothyroidism and Alters Thyroidal Expression of Autophagy Regulators LC3, p62, and ATG5 in Mice"

_ijms, 2025, doi:10.3390/ijms26157660_

Round 1
Reviewer 1 Report
Comments and Suggestions for Authors
Please see the attached file. This is a very intersting and important paper.

Author Response
Reviewer 1
We thank Reviewer 1 for his valuable time and suggestions to improve the manuscript! We have addressed all points as suggested by Reviewer 1, however, due to the major changes in the manuscript, the line numbering might be changed in the current version.
The authors successfully established SPRED3 knockout mice. SPRED3 KO mice showed low body weight, low GH (growth hormone) levels and dysfunction of thyroid (high TSH and low T4 levels) including autophagy. The methods used this study and data analyses are appropriate. This manuscript is well written and contains new and interesting findings.
Only minor editorial revisions are suggested.
Comments 1: Line 196: add comma after amygdala
Response 1: We added comma in line 196.
Comments 2: Line 348: in vitro should be italic
Response 2: We put in vitro in italic.
Comments 3: Line 388: Jiang et al. (2016) should be Jiang et al. [32]
Response 3: We changed the reference in line 388.
Comments 4: Line 403: add commas before and after therefore
Response 4: We added commas in line 403.
Comments 5: Line 415: add space after mice
Response 5: We added space after mice in line 415.
Comments 6: Line 458: add city before CA
Response 6: We added San Mateo before CA in line 458.
Comments 7: Line 478: add city before CA
Response 7: We added Foster City before CA in line 478.
Comments 8: Line 487: add city before MA
Response 8: We added Waltham before MA in line 487.
Comments 9: Line 491: add city before MA
Response 9: We added Cytiva before MA in line 491.
Comments 10: Line 496: add city before MA
Response 10: We added Danvers before MA in line 496.
Comments 11: Line 501: add city before PA
Response 11: We added West Grove before PA in line 501.
Comments 12: Line 516: add city before , Taiwan
Response 12: We added Taipei City before Taiwan in line 516.
Comments 13: Line 517: GB should be UK
Response 13: We changed GB in UK in line 517.
Comments 14: Line 534: delete Sydney, Australia
Response 14: We deleted Sydney, Australia in line 534.
Comments 15: References: need to check all Journal names: need to add a period for the abbreviated word
Response 15: Formatting of references was automatically done with Endnote according to the formatting guidelines of the journal.
Reviewer 2 Report
Comments and Suggestions for Authors
This paper studies the effects of SPRED3 gene deficiency in mice, generating SPRED3 KO mice to investigate the role of this gene in physical development and hormone regulation. SPRED3-deficient mice exhibit growth retardation and the development of hypothyroidism, manifested by elevated TSH and decreased T4 levels. Alterations in autophagy regulatory proteins were also observed, suggesting that SPRED3 is involved in autophagic processes in the thyroid gland. This study offers valuable insights into the physiological significance of SPRED3 and its potential regulatory role in thyroid health, particularly.
However, we would like to make a few suggestions regarding the title of the paper.
1. If you mention autophagy regulators, I think you should at least have data on mTOR, AMPK, transcription factors like TFEB, FOXO, and NFκB, and other factors such as nutrient signals, stress responses (like ER stress and ROS), etc. Therefore, I think it would be better to omit autophagy regulators from the title and add only p62 and ATG5.
2. Regarding the above suggestion, the authors state that the lack of LC3-I/LC3-II resolution in thyroid lysates precluded a conclusive determination of autophagic flux. The authors also stated that effective proteolysis relies not only on autophagosome formation but also on intact transport and fusion mechanisms, which may be impaired in the absence of SPRED3, interpreting reduced p62 levels as ambiguous.
If so, I would like to see them explain if there are any safeguards or countermeasures against this.
3. There seems to be a lack of understanding of the exact molecular mechanism.
3-1. The comprehensive molecular mechanism of MAPK inhibition by SPRED3 is still not fully understood, and it is unclear whether SPRED3 interacts with unique and unknown upstream effectors or regulates alternative feedback loops.
3-2. The detailed molecular mechanisms of hypothyroidism in mice associated with SPRED3 deficiency require further elucidation, particularly concerning autophagosome maturation and turnover.
3-3. The hypothesis that upregulation of SPRED1 compensates for SPRED3 deficiency, contributing to reduced MAPK cascade activation, requires further investigation for a more thorough understanding.
Author Response
Reviewer 2
We thank Reviewer 2 for his valuable time and suggestions to improve the manuscript! We have addressed all points as suggested by Reviewer 1, however, due to the major changes in the manuscript, the line numbering might be changed in the current version.
This paper studies the effects of SPRED3 gene deficiency in mice, generating SPRED3 KO mice to investigate the role of this gene in physical development and hormone regulation. SPRED3-deficient mice exhibit growth retardation and the development of hypothyroidism, manifested by elevated TSH and decreased T4 levels. Alterations in autophagy regulatory proteins were also observed, suggesting that SPRED3 is involved in autophagic processes in the thyroid gland. This study offers valuable insights into the physiological significance of SPRED3 and its potential regulatory role in thyroid health, particularly.
However, we would like to make a few suggestions regarding the title of the paper.
Comments 1: If you mention autophagy regulators, I think you should at least have data on mTOR, AMPK, transcription factors like TFEB, FOXO, and NFκB, and other factors such as nutrient signals, stress responses (like ER stress and ROS), etc. Therefore, I think it would be better to omit autophagy regulators from the title and add only p62 and ATG5.
Response 1: Reviewer 2 is right. Of course one might investigate further markers of autophagic flux. However, we focused on possible interaction partners of SPRED3. But we included the suggestion in the title.
Comments 2: Regarding the above suggestion, the authors state that the lack of LC3-I/LC3-II resolution in thyroid lysates precluded a conclusive determination of autophagic flux. The authors also stated that effective proteolysis relies not only on autophagosome formation but also on intact transport and fusion mechanisms, which may be impaired in the absence of SPRED3, interpreting reduced p62 levels as ambiguous.
If so, I would like to see them explain if there are any safeguards or countermeasures against this.
Response 2: To gain a deeper insight into possibly disrupted processes of autophagy, we compared the LC3-II/I ratio, pBeclin/Beclin ratio, p62, and ATG5 levels in thyroid lysates of KO and WT mice and included the data in the revised version of the manuscript.
Comments 3-1: The comprehensive molecular mechanism of MAPK inhibition by SPRED3 is still not fully understood, and it is unclear whether SPRED3 interacts with unique and unknown upstream effectors or regulates alternative feedback loops.
Response 3-1: This is true, the molecular mechanism is still not fully understood. Therefore we provided data supporting a compensation of SPRED3 loss by SPRED1 upregulation, which might in turn downregulate MAPK signaling (new Figure 6).
Comments 3-2: The detailed molecular mechanisms of hypothyroidism in mice associated with SPRED3 deficiency require further elucidation, particularly concerning autophagosome maturation and turnover.
Response 3-2: We extended the data concerning markers of autophagy and, as stated also above, investigated the LC3-II/I ratio, pBeclin/Beclin ratio, p62, and ATG5 levels in thyroid lysates of KO and WT mice and included the data in the revised version of the manuscript.
Comments 3-3: The hypothesis that upregulation of SPRED1 compensates for SPRED3 deficiency, contributing to reduced MAPK cascade activation, requires further investigation for a more thorough understanding
Response 3-3: We thank the reviewer for this valuable comment. We agree that the compensatory role of SPRED1 in the context of SPRED3 deficiency is an important aspect and warrants further investigation. However, our current study primarily aimed to characterize the phenotypic and molecular consequences of SPRED3 loss. While the observed upregulation of SPRED1 and the corresponding MAPK signaling changes are suggestive of a compensatory mechanism, we believe that a detailed functional validation of this hypothesis lies beyond the scope of the present work. We have clarified this point in the revised manuscript and explicitly state that future studies will be necessary to dissect the specific contributions of SPRED1 to the observed phenotype (see Discussion, lines 446 - 453).
Reviewer 3 Report
Comments and Suggestions for Authors
The study provides valuable insights into the physiological role of SPRED3 through comprehensive characterization of SPRED3 knockout mice, revealing its involvement in thyroid homeostasis and autophagy regulation. However, the novelty of these findings would be strengthened by a more direct comparison with existing literature on SPRED1 and SPRED2, particularly regarding their roles in MAPK signaling and endocrine regulation. Are the observed effects unique to SPRED3, or do they overlap with its paralogs? Clarify how SPRED3’s functions differ or converge with those of other SPRED proteins.
The hypothyroid phenotype in SPRED3 KO mice is compelling, but its translational relevance to human disease is unclear. The manuscript briefly mentions a heterozygous SPRED3 mutation in a human patient (Rosina et al., 2024)—does this individual exhibit thyroid dysfunction? If patient data are unavailable, discuss how future clinical studies could address this gap.
The study reports altered autophagy-related protein expression (e.g., p62, ATG5) in SPRED3-deficient thyroids but does not firmly establish causality between SPRED3 loss and hypothyroidism. To strengthen the mechanistic link:
Include functional assays (e.g., LC3-II/LC3-I ratio, lysosomal activity with LysoTracker) to confirm autophagic flux disruption.
Assess thyroglobulin processing or colloid degradation in thyroid follicles (e.g., by electron microscopy or immunohistochemistry).
The compensatory upregulation of SPRED1 is intriguing but raises questions: Does SPRED1 overexpression alone explain the reduced ERK signaling, or are other feedback mechanisms involved? Consider co-IP or proximity ligation assays to explore SPRED3-SPRED1 interactions.
Author Response
Reviewer 3
The study provides valuable insights into the physiological role of SPRED3 through comprehensive characterization of SPRED3 knockout mice, revealing its involvement in thyroid homeostasis and autophagy regulation.
Comments 1: However, the novelty of these findings would be strengthened by a more direct comparison with existing literature on SPRED1 and SPRED2, particularly regarding their roles in MAPK signaling and endocrine regulation. Are the observed effects unique to SPRED3, or do they overlap with its paralogs? Clarify how SPRED3’s functions differ or converge with those of other SPRED proteins.
Response 1: Thank you very much for this valuable suggestion! We have extended this paragraph in the introduction accordingly.
Comments 2: The hypothyroid phenotype in SPRED3 KO mice is compelling, but its translational relevance to human disease is unclear. The manuscript briefly mentions a heterozygous SPRED3 mutation in a human patient (Rosina et al., 2024)—does this individual exhibit thyroid dysfunction? If patient data are unavailable, discuss how future clinical studies could address this gap.
Response 2: We are very sorry for this but further patient data is unavailable for us. We have tried to contact the authors but did not obtain a response or further data.
Comments 3: The study reports altered autophagy-related protein expression (e.g., p62, ATG5) in SPRED3-deficient thyroids but does not firmly establish causality between SPRED3 loss and hypothyroidism. To strengthen the mechanistic link:
Comments 3-1: Include functional assays (e.g., LC3-II/LC3-I ratio, lysosomal activity with LysoTracker) to confirm autophagic flux disruption.
Response 3-1: We extended the data concerning markers of autophagy and, as stated also above, investigated the LC3-II/I ratio, pBeclin/Beclin ratio, p62, and ATG5 levels in thyroid lysates of KO and WT mice and included the data in the revised version of the manuscript.
Comments 3-2: Assess thyroglobulin processing or colloid degradation in thyroid follicles (e.g., by electron microscopy or immunohistochemistry).
Response 3-2: To address this point, we have compared colloid areas of both groups and found the colloid areas enlarged in KO mice.
Comments 3-3: The compensatory upregulation of SPRED1 is intriguing but raises questions: Does SPRED1 overexpression alone explain the reduced ERK signaling, or are other feedback mechanisms involved? Consider co-IP or proximity ligation assays to explore SPRED3-SPRED1 interactions.
Response 3-3: We thank the reviewer for this valuable comment. We agree that the compensatory role of SPRED1 in the context of SPRED3 deficiency is an important aspect and warrants further investigation. However, our current study primarily aimed to characterize the phenotypic and molecular consequences of SPRED3 loss. Nevertheless, we have tested a potential interaction by computational analyses and included the data in the manuscript. Unfortunately, we do not have access to SPRED1-deficient mice to investigate a possible in vivo compensation. While the observed upregulation of SPRED1 and the corresponding MAPK signaling changes are suggestive of a compensatory mechanism, we believe that a detailed functional validation of this hypothesis lies beyond the scope of the present work. We have clarified this point in the revised manuscript and explicitly state that future studies will be necessary to dissect the specific contributions of SPRED1 to the observed phenotype (see Discussion, lines 446 - 453).
Reviewer 4 Report
Comments and Suggestions for Authors
The manuscript is nicely written with clear objectives and data presentation. Minor comments are suggested:
Were all the analyses performed using adult mice? In other words, did you follow the changes induced by SPRED3 deletion throughout neonatal, pre-adult, and adult stages or only focus on adults? Refer to this in the abstract and when referring to paper limitations as well as future research directions.
Line 24: Specify what is meant by the word “cortex”
Line 27: Specify the trend of changes in autophagy-related proteins.
Figure 1c: Heterozygous mice were used as positive controls for genotyping experiments. Is this scientifically reliable? I understand it has the two alleles of two differing weights, but still not convincing.
The compensation of SPRED1 for SPRED3 loss is interesting. It seems that SPRED3 is partially dispensable based on changes in growth rate and thyroid functions.
Line 448: SPRED3 WT mice: change to “WT mice”
Line 456: Name the tissues and cells used for X-Gal staining.
Line 466: Did you use the whole brain or a particular area?
Lines 494-495: Revise the use of parentheses.
Line 509: Examination of serum hormone levels.
Line 511: … via COâ‚‚ inhalation.
Line 514: Delete the word “supernatant”
Line 538: Provide a reference for AlphaFold4.
Author Response
Reviewer 4:
The manuscript is nicely written with clear objectives and data presentation. Minor comments are suggested:
Comments 1: Were all the analyses performed using adult mice? In other words, did you follow the changes induced by SPRED3 deletion throughout neonatal, pre-adult, and adult stages or only focus on adults? Refer to this in the abstract and when referring to paper limitations as well as future research directions.
Response 1: All experiments were done with organs from adult mice at the age of 8 to 12 months. This was included in the manuscript.
Comments 2: Line 24: Specify what is meant by the word “cortex”
Response 2: We added the word “cerebral” to specify the word “cortex”. We saw the staining in the cerebral cortex of the brain of mice.
Comments 3; Line 27: Specify the trend of changes in autophagy-related proteins.
Response 3: We extended the data concerning markers of autophagy and investigated the LC3-II/I ratio, pBeclin/Beclin ratio, p62, and ATG5 levels in thyroid lysates of KO and WT mice and included the data in the revised version of the manuscript.
Comments 4: Figure 1c: Heterozygous mice were used as positive controls for genotyping experiments. Is this scientifically reliable? I understand it has the two alleles of two differing weights, but still not convincing.
Response 4: One reverse primer (pr. 2) detects the KO allele by binding to a sequence in the gene targeting cassette, the other reverse primer binds to the WT allele. As a result, the upper band represents the KO allele, the lower band the WT allele. If one mouse has two KO alleles, only the upper band will appear, if one mouse has two WT alleles only the lower band will appear, and If one mouse has one KO and one WT allele both bands will appear on the gel. We have extended tihis part in the methods section.
Comments 5: The compensation of SPRED1 for SPRED3 loss is interesting. It seems that SPRED3 is partially dispensable based on changes in growth rate and thyroid functions.
Response 5: We thank the reviewer for this valuable comment. We agree that the compensatory role of SPRED1 in the context of SPRED3 deficiency is an important aspect and warrants further investigation. However, our current study primarily aimed to characterize the phenotypic and molecular consequences of SPRED3 loss. Nevertheless, we have tested a potential interaction by computational analyses and included the data in the manuscript. Unfortunately, we do not have access to SPRED1-deficient mice to investigate a possible in vivo compensation. While the observed upregulation of SPRED1 and the corresponding MAPK signaling changes are suggestive of a compensatory mechanism, we believe that a detailed functional validation of this hypothesis lies beyond the scope of the present work. We have clarified this point in the revised manuscript and explicitly state that future studies will be necessary to dissect the specific contributions of SPRED1 to the observed phenotype (see Discussion, lines 446 - 453).
Comments 6: Line 448: SPRED3 WT mice: change to “WT mice”
Response 6: We changed it to WT mice in line 448.
Comments 7: Line 456: Name the tissues and cells used for X-Gal staining.
Response 7: We have added the names accordingly.
Comments 8: Line 466: Did you use the whole brain or a particular area?
Response 8: For experiments requiring larger amounts of protein and a high SPRED3 expression (e.g. Co-Ips), we have used brain cortex lysates and included this information now in the text.
Comments 9: Lines 494-495: Revise the use of parentheses.
Response 9: We corrected the use of parentheses in lines 494-495.
Comments 10: Line 509: Examination of serum hormone levels.
Response 10: We changed the line 509 according to your suggestion.
Comments 11: Line 511: … via COâ‚‚ inhalation.
Response 11: We added the word inhalation in line 511.
Comments 12: Line 514: Delete the word “supernatant”
Response 12: We deleted “supernantant” in line 12.
Comments 13: Line 538: Provide a reference for AlphaFold4.
Response 13: Reference 44 was added.
Round 2
Reviewer 3 Report
Comments and Suggestions for Authors
This study provides compelling evidence that SPRED3 plays a previously unrecognized role in thyroid homeostasis and autophagy regulation. The generation and characterization of SPRED3 KO mice offer valuable insights into the protein’s endocrine functions, particularly its association with primary hypothyroidism and altered autophagy markers (LC3-II/I, p62, ATG5). The computational prediction of SPRED3 interactions with autophagy regulators (LC3, p62, ATG5) is innovative and supports the experimental findings. This work significantly advances our understanding of SPRED3 beyond its oncogenic roles and justifies publication pending minor clarifications.
- Given the ambiguity in LC3-II/I interpretation, additional experiments (e.g., bafilomycin A1 treatment or lysosomal inhibition assays) would strengthen claims about autophagic flux. At minimum, the text should explicitly acknowledge this limitation.
- The compensatory upregulation of SPRED1 in KO mice is intriguing but requires deeper mechanistic exploration. A brief discussion on whether SPRED1 overexpression alone can explain the reduced p-ERK/ERK ratio would be helpful.
- Clarify the sample sizes for Western blot analyses (e.g., LC3: "WT n=150, KO n=59" in Figure 7 seems atypical; verify if these are technical replicates or biological repeats).
Author Response
First of all, we would like to thank Reviewer 3 for his valuable time and comments! We have addressed all points raised by reviewer 3 and included the changes in the new version of the manuscript.
Reviewer 3:
This study provides compelling evidence that SPRED3 plays a previously unrecognized role in thyroid homeostasis and autophagy regulation. The generation and characterization of SPRED3 KO mice offer valuable insights into the protein’s endocrine functions, particularly its association with primary hypothyroidism and altered autophagy markers (LC3-II/I, p62, ATG5). The computational prediction of SPRED3 interactions with autophagy regulators (LC3, p62, ATG5) is innovative and supports the experimental findings. This work significantly advances our understanding of SPRED3 beyond its oncogenic roles and justifies publication pending minor clarifications.
Comments 1: Given the ambiguity in LC3-II/I interpretation, additional experiments (e.g., bafilomycin A1 treatment or lysosomal inhibition assays) would strengthen claims about autophagic flux. At minimum, the text should explicitly acknowledge this limitation.
Response 1: Reviewer 3 is correct, so far it was not possible to determine in which step the autophagic flux might be disturbed. Therefore, we extended this point in the discussion and included the suggestions of reviewer 3 in lines 496 – 503.
Comments 2: The compensatory upregulation of SPRED1 in KO mice is intriguing but requires deeper mechanistic exploration. A brief discussion on whether SPRED1 overexpression alone can explain the reduced p-ERK/ERK ratio would be helpful.
Response 2: We thank the reviewer for this valuable comment. We agree that the compensatory role of SPRED1 in the context of SPRED3 deficiency is an important aspect and warrants further investigation. However, our current study primarily aimed to characterize the phenotypic and molecular consequences of SPRED3 loss. While the observed upregulation of SPRED1 and the corresponding MAPK signaling changes are suggestive of a compensatory mechanism, we believe that a detailed functional validation of this hypothesis lies beyond the scope of the present work, which focusses on the physiological role of SPRED3. We have clarified this point in the revised manuscript and explicitly state that future studies will be necessary to dissect the specific contributions of SPRED1 to the observed phenotype (see Discussion, lines 446 - 453).
Comments 3: Clarify the sample sizes for Western blot analyses (e.g., LC3: "WT n=150, KO n=59" in Figure 7 seems atypical; verify if these are technical replicates or biological repeats).
Response 3: Thank you very much for the identification of these typographical errors resulting from major changes in the legend of figure 7. This minor error slipped in due to the previous corrections. We have now corrected the number of mice. In the representative Western (7b) are 5 WT and 5 KO displayed, and in the corresponding graph 15 WT and 15 KO evaluable values were included in each group, as also shown by the dots in figure 7c.